# FRCaMP, a Red Fluorescent Genetically Encoded Calcium Indicator Based on Calmodulin from Schizosaccharomyces Pombe Fungus

**DOI:** 10.3390/ijms22010111

**Published:** 2020-12-24

**Authors:** Oksana M. Subach, Natalia V. Barykina, Elizaveta S. Chefanova, Anna V. Vlaskina, Vladimir P. Sotskov, Olga I. Ivashkina, Konstantin V. Anokhin, Fedor V. Subach

**Affiliations:** 1Complex of NBICS Technologies, National Research Center “Kurchatov Institute”, 123182 Moscow, Russia; Subach_OM@nrcki.ru (O.M.S.); Vlaskina_AV@nrcki.ru (A.V.V.); ivashkina_oi@nrcki.ru (O.I.I.); 2Laboratory for Neurobiology of Memory, P.K. Anokhin Research Institute of Normal Physiology, 125315 Moscow, Russia; n.barykina@nphys.ru; 3Department of NBIC-Technologies, Moscow Institute of Physics and Technology, 123182 Moscow, Russia; yelizaveta.chefanova@phystech.edu; 4Institute for Advanced Brain Studies, Lomonosov Moscow State University, 119991 Moscow, Russia; vsotskov@list.ru

**Keywords:** genetically encoded calcium indicator, protein engineering, calcium imaging, *Schizosaccharomyces pombe*, FRCaMP, GECI, red fluorescent, fluorescent protein, split, bJun/bFos

## Abstract

Red fluorescent genetically encoded calcium indicators (GECIs) have expanded the available pallet of colors used for the visualization of neuronal calcium activity in vivo. However, their calcium-binding domain is restricted by calmodulin from metazoans. In this study, we developed red GECI, called FRCaMP, using calmodulin (CaM) from *Schizosaccharomyces pombe* fungus as a calcium binding domain. Compared to the R-GECO1 indicator in vitro, the purified protein FRCaMP had similar spectral characteristics, brightness, and pH stability but a 1.3-fold lower ΔF/F calcium response and 2.6-fold tighter calcium affinity with K_d_ of 441 nM and 2.4–6.6-fold lower photostability. In the cytosol of cultured HeLa cells, FRCaMP visualized calcium transients with a ΔF/F dynamic range of 5.6, which was similar to that of R-GECO1. FRCaMP robustly visualized the spontaneous activity of neuronal cultures and had a similar ΔF/F dynamic range of 1.7 but 2.1-fold faster decay kinetics vs. NCaMP7. On electrically stimulated cultured neurons, FRCaMP demonstrated 1.8-fold faster decay kinetics and 1.7-fold lower ΔF/F values per one action potential of 0.23 compared to the NCaMP7 indicator. The fungus-originating CaM of the FRCaMP indicator version with a deleted M13-like peptide did not interact with the cytosolic environment of the HeLa cells in contrast to the metazoa-originating CaM of the similarly truncated version of the GCaMP6s indicator with a deleted M13-like peptide. Finally, we generated a split version of the FRCaMP indicator, which allowed the simultaneous detection of calcium transients and the heterodimerization of bJun/bFos interacting proteins in the nuclei of HeLa cells with a ΔF/F dynamic range of 9.4 and a contrast of 2.3–3.5, respectively.

## 1. Introduction

Genetically encoded calcium indicators (GECIs) are indispensable tools for the visualization of calcium dynamics in living cells [1,2]. Except for FGCaMP ratiometric green GECI and its enhanced version, FGCaMP7, the calcium-binding parts of GECIs are limited to calmodulin or troponin C from metazoa [3,4]. All popular versions of red GECIs, such as R-GECO1 [5], K-GECO1 [6], and jRGECO1a [7], also utilize the calmodulin (CaM) from metazoa. Because of calmodulin’s lower amino acid sequence homology to metazoa orthologues, the utilization of calmodulin from *Aspergillus niger* fungus in FGCaMP and FGCaMP7 GECIs prevents the interaction of their calcium-binding domains with the intracellular environment [3,4].

Inspired by the success in engineering green GECI using calmodulin from *Aspergillus niger* fungus [3,4], we designed and optimized red GECI for neuronal expression using calmodulin from another fungus, *Schizosaccharomyces pombe*. The developed GECI, called FRCaMP, was characterized in vitro as a purified protein in both the apo- and sat-states. In the cytosol of HeLa cells, FRCaMP showed a ΔF/F response to calcium transients similar to that of R-GECO1 GECI. In neuronal cultures, FRCaMP demonstrated faster calcium dynamics but lower sensitivity compared to NCaMP7 GECI. In contrast to the GCaMP6s GECI, the FRCaMP version with a deleted M13-like peptide did not show any interactions with the intracellular environment in the cytosol of the HeLa cells. Ultimately, in the nuclei of HeLa cells, the split version of the FRCaMP GECI demonstrated greater brightness when fused to the bJun/bFos interacting proteins compared to its fusion with non-interacting control proteins and preserved high response to the calcium transients.

## 2. Results and Discussion

### 2.1. Developing a Red Calcium Indicator Based on cpmApple and CaM and the M13-Like Peptide from S. pombe Fungus in a Bacterial System

The amino acid sequence of calmodulin (CaM) from *S. pombe* is 75% identical to those of CaMs from *H. sapiens* and *M. musculus* (Figure 1a). The percentage of identity among M13-like peptides from CaM-dependent myosin light chain kinases between the same species was below 30% (Figure 1a). These identity percentage values were lower compared to the analogous values for the CaM (85% amino acids identity, ident.) and M13-like peptides (40% ident.) from *A. niger* and *A. fumigatus* fungi that were previously used to develop the FGCaMP green calcium indicator [4]. A circularly permutated version of the mApple red fluorescent protein (cpmApple) from the R-GECO1 indicator was utilized as the fluorescent domain for the original library construction.

First, we generated and analyzed the library with two randomized linkers (each three amino acids in length) between the M13-like peptide and fluorescent part and between the fluorescent and calmodulin parts (Figure 1b). After screening the library on Petri dishes under a fluorescent stereomicroscope followed by an analysis of bacterial lysates on a Plate reader in a 96-well format as described earlier [8], we found a variant with ΔF/F in the presence of 1 mM Mg^2+^ of 2.7 ± 0.5 and a K_d_ calcium affinity constant in the presence of 1 mM Mg^2+^ of 6.7 ± 0.9 nM. To increase its ΔF/F dynamic range and decrease its calcium affinity, this variant was further subjected to five rounds of random mutagenesis followed by screening on Petri dishes and bacterial lysates, as described above. After the fifth round, the selected variant had a response to calcium ions of 5 ± 1 and a K_d_ of 17 ± 3 nM (in the presence of 1 mM Mg^2+^). To adjust its calcium affinity (in the absence of Mg^2+^ ions) to a ~100–200 nM value, which is optimal for the visualization of neuronal activity, we generated and analyzed three libraries with randomized 4, 5, and 6 (library 4–6); 7, 8, and 9 (library 7–9); or 10, 11, and 12 (library 10–12) positions in the M13-peptide (Figure 1b). We found variants with decreased calcium affinities (in the presence of 1 mM Mg^2+^) of 79 ± 4 nM (ΔF/F of 4.8) in library 4–6, 128 ± 6 nM (ΔF/F of 9.9) in library 7–9, and 116 ± 4 nM (ΔF/F of 6.5) and 202 ± 12 nM (ΔF/F of 4.5) in library 10–12. Hence, the mutations in the M13-like peptide affected the calcium affinity and dynamic ranges of the indicator similar to the effects described earlier for the FGCaMP indicator with an M13-like peptide and calmodulin from other fungi—*Aspergillus fumigatus* and *Aspergillus niger*, respectively [4]. The mutants from libraries 7–9 and 10–12 were further subjected to nine rounds of random mutagenesis and screening, as described above.

The final mutant found in the sixteenth round of molecular evolution in the bacterial system was named FRCaMP (Fungus based Red CalModulin/M13-like Peptide). FRCaMP had a design similar to that of R-GECO1 (Figure 1c). FRCaMP had 29 mutations compared to the original library (Figure 1b and Appendix A). Thirteen, five, seven, and four mutations were located in the fluorescent domain, linkers, calmodulin, and M13-like peptide, respectively. Except for two mutations (K262E and X269S), the side chains of the amino acids for all mutations in the fluorescent domain were directed out of the β-barrel; these mutations should not affect the properties of the chromophore (indeed, the in vitro characterization of the FRCaMP’s spectral characteristics described below revealed similar properties compared to the R-GECO1 indicator). We speculate that K262E and X269S mutations can somehow affect the photostability of FRCaMP (as demonstrated below) and these mutations can be considered as the main targets for directed mutagenesis of the FRCaMP indicator in order to improve its photostability; indeed, position 269 was responsible for reversibly photoswitchable-like phenotype in such red fluorescent proteins as rsTagRFP, rsCherry, and KFP1 [9]. Mutations in linkers may dramatically affect the affinity of the FRCaMP indicator to calcium ions [4]. Except for the N370S mutation, all other mutations in the calmodulin domain did not alter the side chains of the putative calcium-binding residues and, therefore, did not dramatically affect the calcium-binding properties of the CaM.

### 2.2. In Vitro Characterization of the Purified FRCaMP Indicator

First, we characterized the spectral and biophysical properties of the purified FRCaMP protein. In the apo- and sat-states, FRCaMP had absorption maxima at 446 and 566 nm, respectively (Figure 2a, and Table 1). These maxima are similar to those for R-GECO1. In the apo-state, FRCaMP was practically non-fluorescent at 446 nm excitation, but the residual absorption peak at 575 nm was dimly fluorescent with excitation/emission maxima at 576/602 nm, similar to R-GECO1 (Figure 2b, and Table 1). In the sat-state, FRCaMP had excitation/emission maxima at 564/592 nm, similar to R-GECO1 (Table 1, and Figure 2b). The molecular brightness of FRCaMP in the sat-state, defined as a product of the quantum yield and extinction coefficient, was 36% of the EGFP brightness, similar to R-GECO1 (Table 1). In the apo-state, the molecular brightness of the FRCaMP at 576 nm excitation was reduced by 14.4-fold as a consequence of the 1.7- and 8.2-fold reduction in the quantum yield and extinction coefficient, respectively (Table 1). Hence, the spectral properties and molecular brightness of the FRCaMP were similar to those of R-GECO1.

For future possible applications of FRCaMP in different cellular organelles, it was important to characterize the impact of pH variation on the dynamic range of the FRCaMP indicator. The dependences of the red fluorescence vs. pH for the FRCaMP indicator in the sat- (pK_a_ of 6.60) and apo-states (pK_a_ of 8.88) (Figure 2c, and Table 1) were similar to the analogous dependences found for the R-GECO1 indicator [5]. Since the pH dependences of the red fluorescence for the sat- and apo-states were different from each other, the ΔF/F dynamic range of the FRCaMP indicator was dramatically dependent on pH values and had a maximal value at a pH of 7.0. The ΔF/F dynamic range of the R-GECO1 indicator was also dramatically affected by pH variations, but the maximum ΔF/F values were observed at a more acidic pH of 6.0 [5]. Hence, similar to the R-GECO1 indicator, the ΔF/F dynamic range of the FRCaMP indicator was strongly dependent on the pH values and was maximal at a pH of 7.0.

In the absence of Mg^2+^ ions, FRCaMP had affinity to calcium ions of 214 ± 6 nM—2.2-fold smaller compared to the calcium affinity of the R-GECO1 indicator (463 ± 10 nM) (Figure 2d, Table 1, and Appendix A). In the presence of 1 mM Mg^2+^ ions (the concentration mimicking that in the cytosol of neurons), FRCaMP had a calcium affinity constant of 441 ± 19 nM—i.e. 2.6-fold smaller than the respective affinity constant of 1138 ± 43 nM for the R-GECO1 indicator (Figure 2d, Table 1, and Appendix A). The addition of 1 mM Mg^2+^ ions shifted the calcium affinity constant of FRCaMP (by 2.1-fold) to a lesser extent than the respective constant of R-GECO1 (by 2.5-fold). Hence, depending on the magnesium ion concentrations, FRCaMP had a 2.2- and 2.6-fold tighter affinity to calcium ions compared to R-GECO1.

Under wide-field one-photon imaging using a mercury lamp, FRCaMP in the sat- and apo-states was photobleached up to 50% 2.4- (*p* = 0.0051) and 6.6-fold (*p* = 0.0079) faster than R-GECO1 in the same states, respectively (Figure 2e,f and Table 1). Hence, under one-photon imaging conditions, FRCaMP had worse photostability than R-GECO1.

mApple0.5 protein revealed notable photochromism [10]. To address the question of whether photochromism affects the photostability of FRCaMP compared to R-GECO1, we photobleached FRCaMP and R-GECO1 GECIs in the sat-state using the mercury lamp described above but with 30-s periods of darkness between the photobleaching cycles (Appendix A). In contrast to R-GECO1, under these conditions FRCaMP demonstrated notable photochromism (Appendix A). Thus, photochromism decreases the one-photon photostability of the FRCaMP indicator.

At a concentration of 11 mg/mL, FRCaMP was eluted as a monomer with only minor signs of the tetramer (Figure 2g). R-GECO1 was crystallized as a dimer [11]. Hence, FRCaMP is a monomer with a weak tendency toward tetramerization. The monomeric state of FRCaMP is advantageous for its possible applications in targeting the indicator and its derivatives to different cellular compartments and in fusion to other proteins.

### 2.3. Calcium-Dependent Response of the FRCaMP Calcium Indicator in HeLa Mammalian Cells

To demonstrate the applicability of the FRCaMP indicator for monitoring calcium transients in the cytosol of mammalian cells, we expressed FRCaMP in the cytosol of the HeLa cells and compared its dynamic range upon the addition of ionomycin to the respective responses for the R-GECO1 indicator. The ΔF/F response of 5.6 ± 2.7 for FRCaMP to ionomycin-induced calcium level elevations in the cytosol of HeLa cells was similar (*p* = 0.0897) to the response of 3.9 ± 1.7 for the R-GECO1 indicator (Table 1 and Figure 3). Hence, FRCaMP is appropriate for monitoring calcium transients in the cytosol of cultured mammalian cells and demonstrates a similar ionomycin-induced ΔF/F response compared to the respective response of the R-GECO1 indicator.

### 2.4. Visualization of Spontaneous Neuronal Activity in the Dissociated Culture Using the FRCaMP Indicator and Confocal Imaging

We next demonstrated the applicability of the FRCaMP indicator for monitoring spontaneous (non-specific) neuronal activity in cultured neuronal cells and closely compared its performance to the green NCaMP7 indicator.

On dissociated neuronal cultures co-expressing red FRCaMP and green NCaMP7 GECIs in the cytosol of the cells, FRCaMP robustly monitored the spontaneous neuronal activity (Figure 4a and Appendix A). Its averaged ΔF/F response of 1.7 ± 1.3 were similar (*p* = 0.1801) to the ΔF/F response of NCaMP7 of 2.9 ± 3.6 (Figure 4b). The average increase (1.8 ± 1.2 s) and decay (5.3 ± 8.5 s) halftimes for FRCaMP were similar (*p* = 0.1801) and 2.1-fold faster (*p* = 0.0001) compared to the rise (1.9 ± 1.3 s) and decay (11.2 ± 11.1 s) halftimes for NCaMP7 (Figure 4c,d).

We noted that after prolonged cultivation, FRCaMP had puncta-like localization in some cells (Appendix A) similar to R-GECO1 [3,6]. Like other green GCaMP6s and FGCaMP7 GECIs [3], green NCaMP7 expressed in the same cells did not reveal puncta-like structures (Appendix A). We suggest that red mApple fluorescent protein (FP)-based indicators are prone to the formation of such puncta due to lysosomal localization [6] driven by autophagy [15]. The lysosomal-autophagy related accumulation is reflecting a normal physiological process of degrading excess of ectopically expressed cytoplasmic proteins and might be regarded as a healthy sign, especially for neurons that are more vulnerable cells in terms of protein aggregation (for instance in Alzheimers, Parkinsons, Huntington, and bovine spongiform encephalopathy (BSE) diseases) [16].

Overall, compared to NCaMP7, FRCaMP robustly monitored spontaneous neuronal activity on the cultured cells, demonstrating similar ΔF/F responses and both similar and 2.1-fold faster rise and decay halftimes, respectively.

### 2.5. Visualization of Induced Neuronal Activity in Dissociated Culture Using the FRCaMP Indicator and Confocal Imaging

To compare the characteristics of the FRCaMP and NCaMP7 indicators in the cultured neurons in more detail, we stimulated the neuronal cultures co-expressing red FRCaMP and green NCaMP7 indicators with an external electric field. According to the stimulation of neuronal cultures with an external electric field, the ΔF/F response of FRCaMP vs. APs was linear in the range of 0–22 APs (Figure 5a–c). On the stimulated neuronal cultures, FRCaMP demonstrated a ΔF/F response per 1 AP of 0.23 ± 0.17 that was 5.6-fold larger, 1.7-fold lower (*p* = 0.0015), and similar (*p* = 0.0880) compared to the respective responses for R-GECO1 (0.04 ± 0.01 [17]), NCaMP7 (0.42 ± 0.09 [18]), and GCaMP6s (0.15 ± 0.08 [3]) indicators (Figure 5c,d). FRCaMP had only 1.5-fold fewer ΔF/F response per 1 AP compared to jRGECO1 [7]. The average rise (1.91 ± 0.69 s) and decay (6.1 ± 3.9 s) halftimes for FRCaMP were similar (*p* = 0.2084) and 1.8-fold faster (*p* = 0.0017) compared to the rise (2.26 ± 0.62 s) and decay (11.1 ± 5.6 s) halftimes for NCaMP7, respectively (Figure 5e,f). Hence, according to the electrical stimulation of the neuronal cultures, FRCaMP demonstrated a 1.7-fold lower ΔF/F response per 1 AP and 1.8-fold faster spike decay dynamics compared to the respective characteristics for the NCaMP7 indicator.

### 2.6. Characterization of Truncated Versions (with a Deleted M13-Like Peptide) of the FRCaMP and GCaMP6s Indicators In Vitro and in HeLa Cells

Since the CaM and M13-like peptide of the FRCaMP indicator had 75% and below 30% amino acid identity with analogous proteins from mammals (Figure 1a), respectively, the FRCaMP indicator might be better than published GECIs based on CaM/M13-like peptide pairs from mammals in terms of its inertness to the intracellular environment. To assess this assumption, we generated a truncated version of the FRCaMP indicator (with a deleted M13-like peptide) and compared its response to calcium ions levels with a similarly truncated version of the popular GCaMP6s indicator both in vitro and in HeLa cells.

The purified truncated versions of FRCaMP and GCaMP6s with deleted M13-like peptide, called FRCaM and GCaM6s, almost did not respond to calcium ions in the range of 0–39, 0–820, and 0–2000 µM free calcium concentrations (Appendix A). This result means that in order to respond to calcium ions, the CaM needs to form a complex with the M13-like peptide in both FRCaMP and GCaMP6s.

The FRCaM and GCaM6s truncated versions had an even distribution when transiently expressed in the cytosol of HeLa cells (Figure 6a,b). The addition of 2.5 µM ionomycin induced an increase in the red fluorescence of FRCaM and green fluorescence of GCaM6s (Figure 6c), with average ΔF/F values of 0.18 ± 0.03 and 1.33 ± 0.28, respectively (Figure 6d). The average ionomycin-induced ΔF/F response of FRCaM in HeLa cells was similar to the ΔF/F response of the purified FRCaM (*p* = 0.6294), but the ΔF/F response of GCaM6s in HeLa cells was 8.1-fold larger compared to the calcium ΔF/F response of the GCaM6s-purified protein (*p* < 0.0001). These results suggest that the CaM in the truncated FRCaM protein does not form a complex with an intracellular protein having an amino acids sequence similar to M13-like peptide or other proteins, and therefore does not interact with the intracellular environment unlike the CaM in the truncated GCaM6s protein. Analogous results were published for similarly truncated versions of the ratiometric green FGCaMP7 indicator, which featured CaM from *Aspergillus niger* fungus and GCaMP6s [3].

Overall, these data indicate that the fungus-originating CaM domain in the FRCaMP indicator does not have partners in the cytosol of mammalian cells in contrast to the metazoa-originating CaM in the GCaMP6s indicator.

### 2.7. Generation and Characterization of the Split-Version of the FRCaMP Indicator in HeLa Cells

Since the truncated version of the FRCaMP indicator with the removed M13-like peptide practically did not respond to the calcium ion transients in the cytosol of HeLa cells, these results prompted us to create and characterize a split-version of the FRCaMP indicator in HeLa cells. The site for the split was chosen between the former N- and C-termini of the circularly permutated fluorescent domain of the FRCaMP indicator. The N- and C-parts of the FRCaMP indicator (FRCaMPN and FRCaMPC, respectively) were fused to the bFos and bJun hetero-dimerizing proteins; as a control, a bFOSΔZip–FRCaMPN fusion, which is not capable of interacting with bJun, was also engineered [19].

We next co-expressed a bJun-FRCaMPC/bFos-FRCaMPN hetero-dimerizing pair and control bJun-FRCaMPC/bFOSΔZip-FRCaMPN non-dimerizing pair in HeLa cells and compared their ΔF/F responses to ionomycin-induced calcium transients and brightness at low physiological and elevated calcium ions concentrations in the nuclei of the cells. The bJun-FRCaMPC/bFos-FRCaMPN and bJun-FRCaMPC/bFOSΔZip-FRCaMPN control pairs demonstrated an uneven distribution in the nuclei of HeLa cells (Figure 7a,b and Appendix A). Compared to the control bJun-FRCaMPC/bFOSΔZip-FRCaMPN pair, upon ionomycin addition, the bJun-FRCaMPC/bFos-FRCaMPN pair demonstrated a 3.53-fold (*p* < 0.0001) larger average brightness at elevated calcium concentrations (Figure 7c) and 1.66-fold larger ΔF/F response (Figure 7c and Appendix A) to the calcium elevation, with average responses of 9.4 ± 2.8 and 5.6 ± 1.9 for the bJun-FRCaMPC/bFos-FRCaMPN and bJun-FRCaMPC/bFOSΔZip-FRCaMPN pairs, respectively. At low physiological calcium ions concentration, bJun-FRCaMPC/bFos-FRCaMPN pair demonstrated 2.3-fold (*p* < 0.0001) larger averaged brightness compared to the control bJun-FRCaMPC/bFOSΔZip-FRCaMPN pair (Appendix A). Hence, the split-version of the FRCaMP indicator allowed the simultaneous visualization of both calcium transients and protein–protein interactions in the nuclei of the mammalian cells with a 9.4 ΔF/F dynamic range and 2.3–3.5-fold contrast, respectively. Overall, in the specific case described above, the spit FRCaMP-based protein–protein interactions sensing system additionally allows the detecting calcium ions elevation in the same fluorescence channel, suggesting that for this reason it has a particular advantage over the regular split FP-based fluorescence complementation system; however, in the general case, we do not recommend using a FRCaMP-based split system for detecting protein–protein interactions due to heterodimerization between M13-like peptide and CaM at the elevated calcium ions concentrations, which can lead to incorrect conclusions about protein–protein interactions.

## 3. Materials and Methods

### 3.1. Mutagenesis and Library Screening

Library construction and subsequent screening were performed as described in [8]. FRCaMP variants were cloned into the pBAD/HisB-TorA-mTagBFP plasmid using the RSP-BglII/CaMSP-HindIII-r primers listed in Appendix A to express FRCaMP variants in the periplasm of bacterial cells. Libraries for the randomization of several positions at the M13-peptide were generated using the MSP-3–5, MSP-6–8, or MSP-9–11 primers listed in Appendix A.

### 3.2. Protein Purification and Characterization

Proteins were expressed and purified as described in [20]. All experiments for protein characterization were performed at room temperature (r.t.).

The extinction coefficient values for the purified FRCaMP protein in a Ca^2+^-saturated state or apo-state were calculated in 30 mM HEPES buffer, pH 7.2 (adjusted to desired pH by using KOH), 100 mM KCl (buffer A), supplemented with 5 mM CaCl_2_ or 10 mM EDTA, respectively, using the alkaline denaturation method and assuming that the mCherry-like red chromophore had an extinction coefficient of 38,000 M^−1^ cm^−1^ at 455 nm in 1 M NaOH [5]. The absorption spectra were registered using a NanoDrop 2000c Spectrophotometer (Thermo Scientific, Wilmington, DE, USA).

To determine the quantum yield of purified FRCaMP excited at 540 nm in a Ca^2+^ saturated state or apo-state, the integrated fluorescence values (in the range of 550–750 nm) for the protein were measured in buffer A supplemented with 5 mM CaCl_2_ or 10 mM EDTA, respectively, and compared with the same values for the equally absorbing at 540 nm mCherry protein (quantum yield of 0.22 [12]). To determine the quantum yield of purified FRCaMP and R-GECO1 proteins excited at 400 nm in the apo-state, the integrated fluorescence values (in the range of 410–750 nm) of the proteins were measured in buffer A supplemented with 10 mM EDTA and compared with the same values for the equal absorption at 400 nm of the mTagBFP2 protein (quantum yield of 0.64 [13]). The fluorescence spectra were acquired using a CM2203 spectrofluorometer (Solar, Minsk, Belarus).

pH titrations for purified FRCaMP protein (50 nM final concentration) in a Ca^2+^-saturated state or apo-state were performed in buffers of 30 mM citric acid, 30 mM borax, and 30 mM NaCl with pH values ranging from 3.0 to 10.5 (adjusted to desired pH by using HCl or NaOH), supplemented with 0.1 mM CaCl_2_ or 0.1 mM EDTA, respectively, followed by incubation for 20 min, as described in [20]. Red fluorescence (Ex525nm/Em580–640 nm) was registered using a 96-well ModulusTM II Microplate Reader (Turner Biosystems, Sunnyvale, CA, USA).

To determine the equilibrium calcium K_d_, the fluorescence of the FRCaMP protein (50 nM final concentration) was measured in a mixture of two stock buffers of 30 mM 3-(N-morpholino)propanesulfonic acid (MOPS), pH 7.2 (adjusted to desired pH by using KOH), and 100 mM KCl (buffer B) containing 10 mM EGTA or 10 mM Ca-EGTA, as described previously [20]. Two buffers, buffer B supplemented with either 10 mM EGTA or 10 mM Ca-EGTA, were mixed to 0, 0.25, 0.5, 1, 2, 3, 4, 5, 6, 7, 8, 9, 9.5, 9.75, and 9, 875, and 10 mM Ca_total_ concentrations to obtain 2.1, 7.9, 16.3, 35.4, 79.2, 135.5, 208.4, 312.6, 468.9, 731.5, 1254.6, 2817.6, 5856.0, 11858.0, 21923.7, and 81,276 nM Ca_free_ concentrations (the free calcium concentrations were corrected by 1.084-fold according to the GCaMP6s indicator K_d_ at 144 nM [21]). To determine the equilibrium calcium K_d_ in the presence of 1 mM Mg^2+^ ions, the fluorescence of the FRCaMP protein (50 nM final concentration) was measured in a mixture of two stock buffers, buffer B supplemented with 1 mM MgCl_2_ and either 10 mM EGTA or 10 mM Ca-EGTA. The dilutions of the stock solutions were the same, and free calcium concentrations were considered as described above in the absence of Mg^2+^ ions. Red fluorescence (Ex525nm/Em580–640nm) was registered using a 96-well ModulusTM II Microplate Reader (Turner Biosystems, Sunnyvale, CA, USA).

Photobleaching experiments were performed with suspensions of purified proteins in mineral oil, as previously described [22]. Briefly, the kinetics of photobleaching were measured using purified proteins dialyzed in buffer A supplemented with either 10 mM EDTA or 5 mM CaCl_2_ at a 92 µM concentration in aqueous microdroplets in mineral oil using a Zeiss Axio Imager Z2 microscope (Zeiss, Germany) equipped with a 120 W mercury short-arc lamp (LEJ, Germany), a 63 × 1.4 NA oil immersion objective lens (PlanApo, Zeiss, Germany), a 550/25BP excitation filter, a FT 570 beam splitter, and 605/70BP emission filters. Light power density (9.136 mW/cm^2^) was measured at the rear focal plane of the objective lens using a PM100D power meter (ThorLabs, Germany) equipped with an S120VS sensor (ThorLabs, Germany). No corrections were applied to the experimental data.

### 3.3. Bacterial and Mammalian Plasmids Construction

To construct the pAAV-*CAG*-*NES-FRCaMP* plasmid, the *FRCaMP* gene was PCR-amplified like the BglII-HindIII fragment using the RSP-BglII/CaMSP-HindIII-r primers listed in Appendix A and swapped with the *mCherry* gene in the pAAV-*CAG*-*NES-mCherry* vector.

To construct the pBAD/HisB-*TorA-FRCaM* and pAAV-*CAG*-*NES-FRCaM* plasmids with the *FRCaMP* version containing the deleted M13-like peptide (called *FRCaM*), the *FRCaM* gene was PCR-amplified like the BglII-HindIII fragment using the dRSP-BglII2/CaMSP-HindIII-r primers listed in Appendix A and swapped with the *mTagBFP2* or *mCherry* genes in the pBAD/HisB-*TorA-mTagBFP2* and pAAV-*CAG*-*NES-mCherry* vectors, respectively.

To construct the pAAV-*CAG*-*bJun-FRCaMPC*, pAAV-*CAG*-*bFos-FRCaMPN*, and pAAV-*CAG*-*bFOSΔZip-FRCaMPN* plasmids, the *FRCaMPC* and *FRCaMP*N genes were PCR-amplified like the NheI-HindIII fragments using the NheI-RSPC/CaMSP-HindIII-r or NheI-RSPN/RSPN-HindIII-r primers listed in the Appendix A and swapped with the *mTagRFP* gene in the pAAV-*CAG*-*bJun-mTagRFP*, pAAV-*CAG*-*bFos-mTagRFP*, and pAAV-*CAG*-*bFOSΔZip-mTagRFP* vectors, respectively.

To construct the pAAV-*CAG*-*bJun-mTagRFP*, pAAV-*CAG*-*bFos-mTagRFP*, and pAAV-*CAG*-*bFOSΔZip-mTagRFP* plasmids, the *bJun*, *bFos*, and *bFOSΔZip* genes were PCR-amplified from pBiFC-*bJunVN173* (Addgene #22012), pBiFC-*bFosVC155* (Addgene #22013), and pBiFC-*bFOSDeltaZipVC155* (Addgene #22014) plasmids like the BamHI-NheI fragments using the BamHI-bJun/bJun-NheI-r or BamHI-bFos/bFos-NheI-r primers listed in Appendix A and inserted into the pAAV-*CAG*-*mTagRFP* plasmids at the BamHI/NheI restriction sites.

### 3.4. Mammalian Live-Cell Imaging

HeLa Kyoto cell cultures were imaged 24–48 h after transient lipofectamine transfection before and immediately after 2.5 μM Ionomycin addition using a laser spinning-disk Andor XDi Technology Revolution multi-point confocal system (Andor Technology, Belfast, UK), as previously described [20]. Then, 20 mM HEPES, pH 7.40 (adjusted to desired pH by using NaOH), was added before imaging and cells were kept at r.t during experiment. To quantify the fluorescence intensity, the background noise determined from the adjacent cell-free area was subtracted from mean fluorescence intensity value for cytosolic (Figure 3 and Figure 6) or nuclei (Figure 7) sub-region of the cell of the similar area.

### 3.5. Imaging in Primary Mouse Neuronal Cultures

The rAAV particles were purified from ten 150 cm dishes, as described in the original paper [8]. Dissociated neuronal cultures were isolated from the C57BL/6 mice at postnatal days 0–1 and were grown on a 24-well cell imaging black plate with a glass bottom; then, the tissue cultures were treated (Eppendorf, Hamburg, Germany) in Neurobasal Medium A (GIBCO, Paisley, Scotland, UK) supplemented with 2% B27 Supplement (GIBCO, Paisley, Scotland, UK), 0.5 mM glutamine (GIBCO, Paisley, Scotland, UK), 50 U/mL penicillin, and 50 µg/mL streptomycin (GIBCO, Paisley, Scotland, UK). On the fourth day in vitro, neuronal cultures were transduced with a mixture of rAAV viral particles (DJ serotype) carrying AAV-*CAG-NES-FRCaMP* and AAV-*CAG-NES-NCaMP7*. The cells were imaged using an Andor XDi Technology Revolution multi-point confocal system on DIV 15 (spontaneous activity at 37 °C, 5% carbon dioxide) and 21–22 (electrical field stimulation at r.t.).

Stimulation of neuronal cultures was performed using a self-built electrical system described earlier [8]. In this step, 300 voltage pulses of a 1 ms duration (0.5 ms negative phase, 0.5 ms interphase, and 0.5 ms positive phase) at a 87 Hz frequency with an amplitude of ±70 V were applied to the neuronal cultures in 24-well plates through iridium electrodes with a 5 mm gap. Then, 10 µM cyanquixaline (6-cyano-7-nitroquinoxaline-2,3-dione) (CNQX) and 100 µM (2R)-amino-5-phosphonovaleric acid (APV or AP5) were added before stimulation to block spontaneous neuronal activity. To quantify the fluorescence intensity, the background noise determined from the adjacent cell-free area was subtracted from mean fluorescence intensity value for the cytosolic (Figure 4 and Figure 5) sub-region of the cell of the similar area. The rise and decay half-times were calculated as time difference between time point corresponding to the calcium spike maximum and time points at half-maximum at the left and right edges of the spike, respectively.

### 3.6. Statistics

To estimate the significance of the difference between two values, we used a Mann–Whitney Rank Sum Test and provided *p*-values (throughout the text in the brackets) calculated for the two-tailed hypothesis. We considered the difference to be significant when the *p*-value was <0.05.

### 3.7. Ethical Approval and Animal Care

All methods for animal care and all experimental protocols were *approved* by the National Research Center “Kurchatov Institute” Committee on Animal Care (NG-1/109PR of 13 February 2020) and were done *in accordance* with the *Russian* Federation Order Requirements N *267* MЗ and the National Institutes of Health Guide for the Care and Use of Laboratory Animals. Two old C57BL/6 P0–1 mice were used in this study. The mice were used without regard to gender.

## 4. Conclusions

In conclusion, we developed a red FRCaMP indicator based on CaM from *S. pombe* fungus and characterized its properties in vitro as a purified protein; compared to the purified R-GECO1 indicator, FRCaMP had similar spectral properties and molecular brightness, a 1.3-fold lower dynamic range, 2.6-fold higher calcium affinity, and 2.4–6.6-fold lower photostability.

In the cytosol of mammalian cells, FRCaMP visualized calcium transients with similar ΔF/F responses to R-GECO1. FRCaMP robustly visualized the spontaneous activity of neuronal cultures with similar ΔF/F response and 2.1-fold faster decay kinetics as directly compared to the analogous characteristics for the NCaMP7 indicator. On electrically stimulated neuronal cultures, FRCaMP demonstrated 1.7-fold lower and 1.8-fold faster ΔF/F responses per 1 AP and decay kinetics, respectively, when directly compared to those for the NCaMP7 indicator; the ΔF/F response per 1 AP for FRCaMP was also 5.6-fold superior, similar to, and 1.5-fold lower compared to the R-GECO1, GCaMP6s, and jRGECO1 [7] indicators, respectively. Some neuronal cells expressing the FRCaMP indicator showed puncta-like structures similar to those of R-GECO1-expressing neurons, so the different origins of the CaM-domain did not help resolve this issue, and the common fluorescent domain of these indicators is likely responsible for this phenomenon.

Using truncated versions of the indicators, we showed the absence of interactions between the FRCaMP fungus-originating CaM and the cytosol environment of the mammalian cells in contrast to the metazoa-originating CaM of GCaMP6s. The FGCaMP7 indicator including CaM from another fungus, *A. niger*, also does not have partners in the cytosol of mammalian cells [3]. Thus, the utilization of CaMs from fungi is a good strategy for developing GECIs and preventing GECIs from interacting with the intracellular environment of mammalian cells. Since the truncated version of the FRCaMP indicator (called FRCaM) practically does not change its fluorescence in response to calcium ions and does not have partners in the cytosol of mammalian cells, this FRCaM can be used as a control to estimate the possible impacts from variations like pH; in the case of the GCaMP6s indicator, this truncated version cannot be used as a control due to its interactions with the cytosolic environment (Figure 6b).

Hence, FRCaMP is a good alternative to the jRGECO1 indicator [7] for both in vitro and in vivo calcium imaging but with less of an impact on cellular properties potentially caused by its overexpression.

We successfully generated a split-version of the FRCaMP indicator, which, in a specific case, was able to visualize simultaneously both calcium transients and protein–protein interactions in the nuclei of HeLa cells with a high ΔF/F dynamic range of 9.4- and contrast of 2.3–3.5-fold, respectively. These data suggest that the elevated fluorescence of the split-version of FRCaMP itself without additional fusion proteins emerges in a calcium concentration-dependent manner due to the presence of the CaM and M13-like peptide proteins interacting at elevated calcium concentrations. Thus, the expression of the split-FRCaMP indicator under a doxycycline-regulated promoter in the time window might be a good alternative to CAMPARI indicator–integrator-based technology [23] for labeling calcium neuronal activity throughout the brain without the need to utilize light illumination. Further efforts are needed to prove and develop this novel type of application for split-FRCaMP GECI.

## Figures and Tables

**Figure 1 ijms-22-00111-f001:**
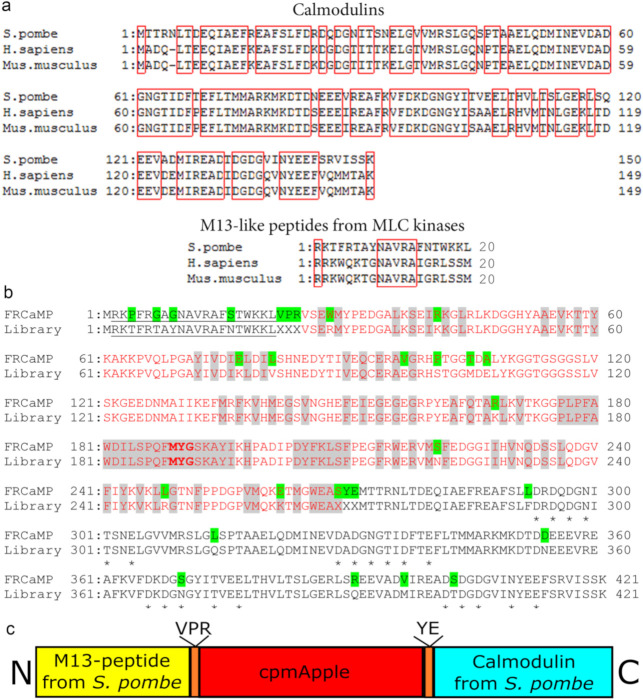
Alignment of the amino acid sequences for calmodulins (CaMs) and M13-like peptides from the CaM-dependent myosin light chain (MLC) kinases found in different species (**a**) and for the FRCaMP indicator and its original library (**b**) and FRCaMP design (**c**). (**a**) Identical residues are highlighted with red boxes. (**b**) Alignment numbering follows that used for the FRCaMP protein. Mutations in FRCaMP relative to the original library, as well as linkers between the fluorescent (in red) and calcium-binding parts, are highlighted in a green color. The residues buried inside the β-barrel of the fluorescent part are highlighted in a grey color. Residues in the CaM-part that are assumed to bind Ca^2+^ ions are highlighted with stars at the bottom. The M13-peptide is underlined. The chromophore-forming residues **MYG** are highlighted in bold. (**c**) The arrow deciphers the amino acid composition of the linkers.

**Figure 2 ijms-22-00111-f002:**
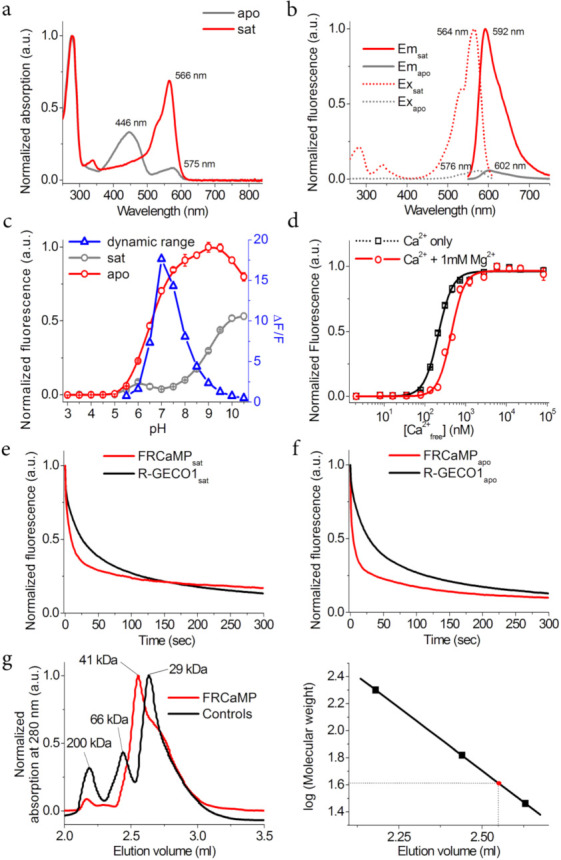
In vitro properties of the purified FRCaMP indicator. (**a**) Absorption spectra for FRCaMP in the Ca^2+^-bound (sat) and Ca^2+^-free (apo) states at pH 7.20. (**b**) Excitation and emission spectra for FRCaMP in the Ca^2+^-bound (sat) and Ca^2+^-free (apo) states at pH 7.20. (**c**) Red fluorescence intensity for FRCaMP in the Ca^2+^-bound (sat) and Ca^2+^-free (sat) states and the ΔF/F dynamic range as a function of pH. (**d**) Ca^2+^ titration curves for FRCaMP in the absence and presence of 1 mM MgCl_2_ at pH 7.20. The experimental data were fitted by the Hill equation. (**e**,**f**) Photobleaching of FRCaMP under continuous wide-field imaging using a mercury lamp in the presence (**e**) or absence (**f**) of Ca^2+^. (**g**) Fast protein liquid chromatography of FRCaMP in the presence of Ca^2+^. FRCaMP (11 mg/mL) was eluted in 40 mM Tris-HCl (pH 7.5) and 200 mM NaCl buffer supplemented with 5 mM CaCl_2_. The molecular weight of FRCaMP (having a theoretical molecular weight of 47 kDa) was calculated from a linear regression of the dependence of the logarithm of the control molecular weights vs. the elution volume. (**c**–**f**) Three–seven replicates were averaged for analysis. Error bars represent the standard deviation.

**Figure 3 ijms-22-00111-f003:**
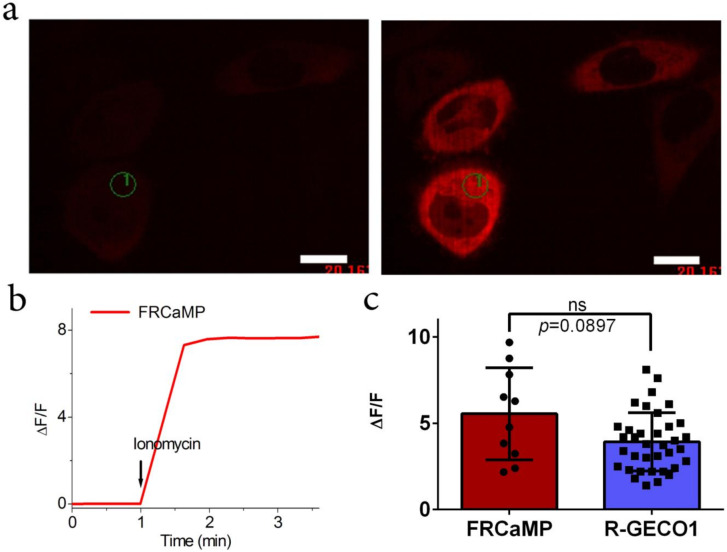
Localization and response of the FRCaMP indicator to Ca^2+^ variations in HeLa cells. (**a**) Confocal images of HeLa cells expressing the FRCaMP calcium indicator before and after the addition of 2.5 μM ionomycin. The red fluorescence channel corresponds to excitation and emissions at 561 and 617/73 nm, respectively. Scale bar, 20 µm. (**b**) The graph illustrates ΔF/F changes in the red fluorescence of the FRCaMP indicator in response to the addition of 2.5 μM ionomycin. The changes on the graph correspond to the area indicated on panel a as a numbered circle. The time of ionomycin addition is shown by an arrow. (**c**) Comparison of the averaged ΔF/F responses for the FRCaMP (two cultures) and R-GECO1 (seven cultures) indicators in HeLa cells upon 2.5 μM ionomycin addition in different cell cultures. Error bars indicate the standard deviations across 10–36 cells. *p*-values show a statistical difference between the respective values. ns, not significant.

**Figure 4 ijms-22-00111-f004:**
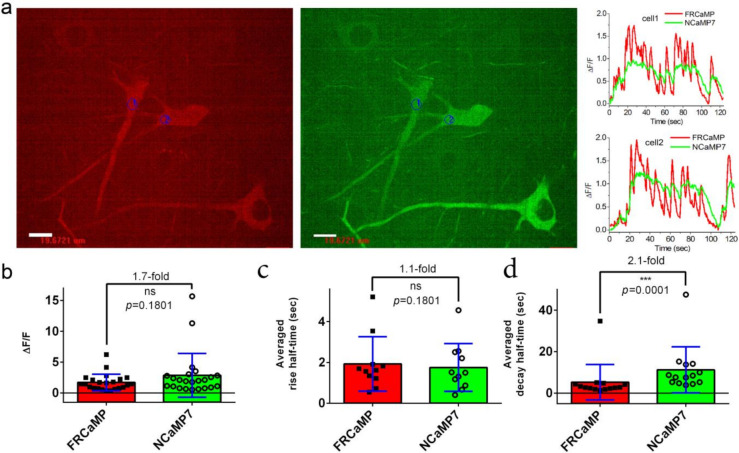
Two-color calcium imaging of the non-specific (spontaneous) activity of neuronal cultures co-expressing the red FRCaMP and green NCaMP7 indicators. Neuronal cultures were transduced on day in vitro (DIV) fourth with a mixture of rAAVs carrying the nuclear export signals (NES)-FRCaMP and NES-NCaMP7. (**a**) Confocal images of neuronal cultures co-expressing the NES-FRCaMP and NES-NCaMP7 indicators. Scale bar, 20 µm. Examples of ΔF/F traces for the two selected cells labeled with circles (see Appendix A for more traces). Neuronal cultures co-expressing the NES-FRCaMP and NES-NCaMP7 indicators were imaged on DIV 15th. Average ΔF/F responses (**b**), rise, and (**c**) decay (**d**) halftimes for the FRCaMP and NCaMP7 indicators. (**b**–**d**) Error bars are the standard deviations across 16 cells and 11–23 spikes. Ns, not significant; *p*, the level of statistical significance; ***, *p*-value is 0.0001 to 0.001.

**Figure 5 ijms-22-00111-f005:**
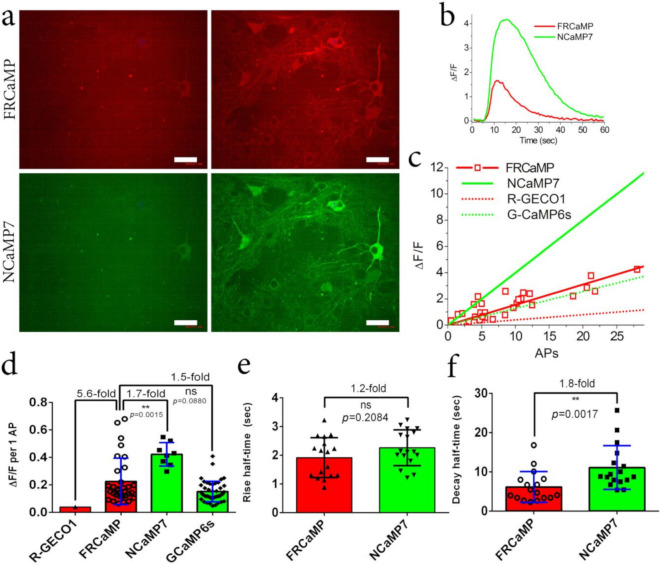
Comparison of the responses of red FRCaMP and green NCaMP7 indicators to the external field stimulation of neurons co-expressing the GECIs in dissociated neuronal cultures. Neuronal cultures co-expressing the NES-FRCaMP and NES-NCaMP7 indicators were imaged and stimulated on DIV 21–22th. Neuronal cultures were transduced on DIV fourth with a mixture of rAAVs carrying NES-FRCaMP and NES-NCaMP7. (**a**) Confocal images of neuronal culture co-expressing the NES-FRCaMP and NES-NCaMP7 indicators before (left) and after (right) electrical stimulation. Scale bar, 50 µm. (**b**) The graph illustrates ΔF/F changes in the red and green fluorescence of the FRCaMP and NCaMP7 indicators in response to an electrical field stimulation. The changes on the graph correspond to the area indicated on panel a as a numbered circle. (**c**) The dependence of ΔF/F responses for the FRCaMP indicator vs. the number of action potentials (APs). A number of APs was determined according to the ΔF/F response of the NCaMP7 indicator (0.4 per 1 AP [18]) co-expressed in the same cell and assuming that response’s linearity in the examined AP range (linear fitting for FRCaMP had R^2^ value of 0.7006). The dependences of ΔF/F responses on APs for R-GECO1 and G-CaMP6s were added to compare the results with previous work [3]. (**d**) The ΔF/F response per one AP for FRCaMP was calculated according to the ΔF/F response of NCaMP7 (0.4 per 1 AP [18]) in the same cell. For comparison, the ΔF/F responses per one AP for R-GECO1 and G-CaMP6s were added from previous work [3]. (**e**,**f**) The rise (**e**) and decay (**f**) halftimes for FRCaMP and NCaMP7. (**d**–**f**) Error bars are the standard deviations across 17–32 cells. Ns, not significant. **, *p*-value is 0.001 to 0.01.

**Figure 6 ijms-22-00111-f006:**
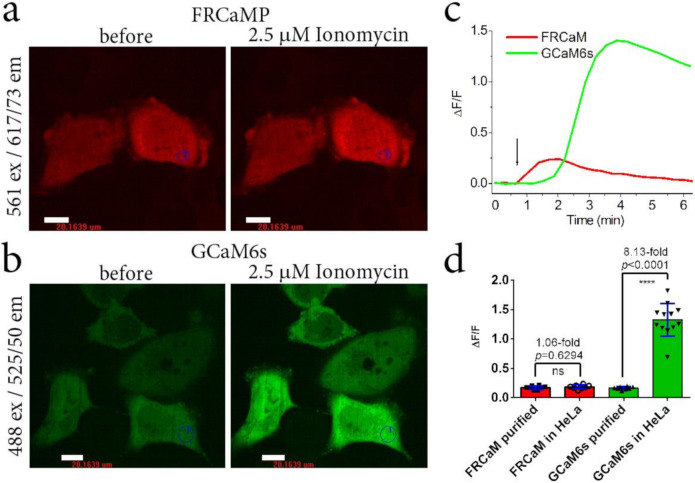
Calcium-dependent response of the truncated versions (with a deleted M13-like peptide) of the FRCaMP (FRCaM) and GCaMP6s (GCaM6s) indicators in HeLa cells. Confocal images of HeLa cells expressing red FRCaM (**a**) and green GCaM6s (**b**) before and after the addition of 2.5 μm ionomycin. (**c**) Graph illustrating the calcium-dependent changes in ΔF/F for FRCaM (red) and GCaM6s (green). The addition of 2.5 μm ionomycin is depicted by a black arrow. (**d**) Average ionomycin-evoked ΔF/F responses of the FRCaM (*n* = 8) and GCaM6s (*n* = 12) truncated indicators in HeLa cells in comparison to the average ΔF/F_0_ responses to the addition of 39 µM calcium for the purified FRCaMP (*n* = 8) and GCaM6s (*n* = 8) truncated indicators.

**Figure 7 ijms-22-00111-f007:**
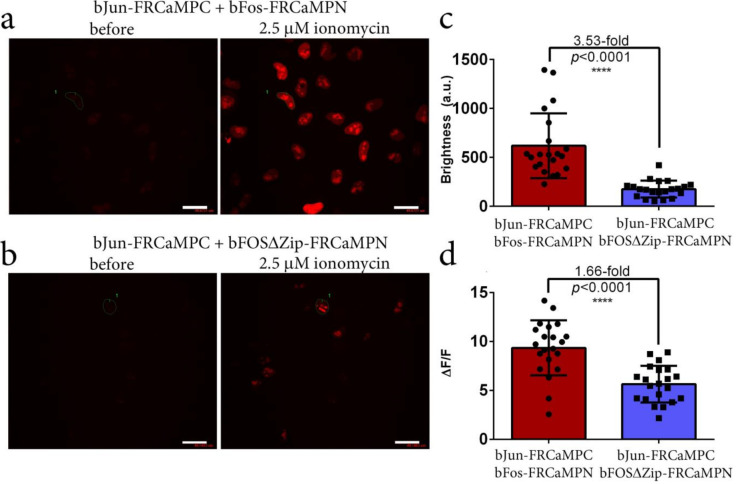
Localization, brightness, and responses of the split-version of the FRCaMP indicator to Ca^2+^ variations in HeLa cells depending on the presence of the heterodimerizing (bJun-FRCaMPC/bFos-FRCaMPN) or non-heterodimerizing (bJun-FRCaMPC/bFOSΔZip-FRCaMPN) pair. (**a**) Confocal images of HeLa cells co-expressing the heterodimerizing bJun-FRCaMPC and bFos-FRCaMPN split calcium indicator before and after the addition of 2.5 μM ionomycin. (**b**) Confocal images of HeLa cells co-expressing the control non-heterodimerizing bJun-FRCaMPC and bFOSΔZip-FRCaMPN split calcium indicator before and after the addition of 2.5 μM ionomycin. (**a**,**b**) Red fluorescence channel corresponding to excitation and emission at 561 and 617/73 nm, respectively. Scale bar, 50 µm. For the selected cells (marked with label 1), zoomed images and time-lapses are shown in Appendix A. The brightness levels are the same for both panels in the comparison. (**c**,**d**) Comparison of the average brightness (**c**) and ΔF/F responses (**d**) for the heterodimerizing bJun-FRCaMPC and bFos-FRCaMPN split calcium indicator and their control non-heterodimerizing bJun-FRCaMPC and bFOSΔZip-FRCaMPN split calcium indicator in HeLa cells upon the addition of 2.5 μM ionomycin. Comparison of the averaged brightness before ionomycin addition at low calcium ions concentrations is shown in Appendix A. Error bars represent the standard deviations across twenty-one cells (three cultures); *p*-values show statistical differences between the respective values. ****, the *p*-value is lower than 0.0001. We used 1600 ng concentration of the plasmids for the transfection in 24-well format.

**Table 1 ijms-22-00111-t001:** In vitro properties of the FRCaMP calcium indicator compared to the R-GECO1 indicator.

Properties			FRCaMP		R-GECO1 ^a^	
			Apo	Sat	Apo	Sat
Absorption maximum (nm)			446 (575)	566	445 (577)	561
Excitation maximum (nm)			NF ^b^ (576)	564	NF * (578) *	564 *
Emission maximum (nm)			NF (602)	592	NF * (600)	589
Quantum yield (%) ^c^			<0.004 (0.133 ± 0.003)	0.228 ± 0.008	<0.006 * (0.06)	0.20
ε (mM^−1^·cm^−1^) ^d^			26.16 ± 0.08 (6.47 ± 0.07)	53 ± 2	22 (15)	51
Brightness vs. EGFP (%) ^e^			0 (2.5)	36	0 * (2.7)	30
p*K*_a_			8.88 ± 0.05	6.60 ± 0.04	8.9	6.59
ΔF/F	Purified protein	0 mM Mg	16.4 ± 0.7		20 ± 1 *	
		1 mM Mg	15.8 ± 0.5		21.0 ± 0.2 *	
	HeLa cells		5.6 ± 2.7		3.9 ± 1.7 *	
K_d_ (nM) ^f^	0 mM Mg		214 ± 6 (2.5 ± 0.2)		463 ± 10 (1.92 ± 0.07) *	
	1 mM Mg		441 ± 19 (2.7 ± 0.3)		1138 ± 43 (2.02 ± 0.12) *	
Monomeric state			monomer		dimer	
Photobleaching halftime (sec) ^g^			4.3 ± 1.4	10.8 ± 5.3	28.5 ± 6.6 *	25.6 ± 4.1 *

^a^ Data from [5,11]. Data marked with an asterisk (*) were determined in this paper. ^b^ NF, non-fluorescent. ^c^ QY was determined at pH 7.20. mCherry (QY = 0.22 [12]) and mTagBFP2 (QY = 0.64 [13]) were used as the reference standards. ^d^ The extinction coefficient for the form with an absorption maximum at 566 nm was determined by alkaline denaturation. ^e^ Brightness was calculated as a product of the quantum yield and extinction coefficient and normalized to the brightness of EGFP, which has an extinction coefficient of 56,000 M^−1^·cm^−1^ and a quantum yield of 0.6 [14]. ^f^ The Hill coefficient is shown in brackets. ^g^ Halftime to bleaching up to 50%. One-photon photobleaching was performed under a mercury lamp with drops in oil. Standard deviations are shown.

## Data Availability

Data is contained within the article or supplementary material.

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
