# Peer review of "FRCaMP, a Red Fluorescent Genetically Encoded Calcium Indicator Based on Calmodulin from Schizosaccharomyces Pombe Fungus"

_ijms, 2020, doi:10.3390/ijms22010111_

Round 1
Reviewer 1 Report
The paper by Subach et al describes the evolution and characterization of a new red fluorescent calcium sensor FRCaMP that incorporates a calcium sensing domain originating from fungus.
Overall, the paper is clear and convincing in describing the evolution and characterization of the new red calcium sensor. Because many aspects are quantified also a good comparison can be made to existing state of the art.
There are four points of attention:
- 1) In the evolution, not only new calcium sensing domains (such as CaM and M13) and linkers were included, but also the circularly permutated mApple RFP was mutagenized. The rationale for the latter is not explained. In the evolution many amino acids of the cpApple were altered. Although the main spectral properties of the cpRFP in the sensor are not changed, the photostability seems to be much decreased. Which seems to be the only drawback of the FRCaMP sensor. The question is whether this can be pinpointed to certain amino acids in the cpApple part, by looking at the diverse mutants that they isolated during evolution? The other question is whether this decreased photostability is also partially explained by photochromism. For instance, it is known that mApple is strongly photochromic (quenched by yellow light and restored with blue light). Hence the question here is to what extent the decreased photostability of FRCaMP vs RCaMP is explained by reversible photochromism and/or irreversible photobleaching ? And to what extent can it be attributed to the mutations found in the cpApple RFP?
- 2) The rationale for developing the split sensor (fig 7) is unclear. First it seems that the negative control designed to not show an interaction does show fluorescence (albeit 3.5 times lower, see fig 7c) and hence the question is: how specific is this? To screen for an unknown interaction this split system would yield many false positives. Furthermore, it seems not to be advantageous to need to use Ionomycin to read out the protein-protein interaction (as compared to a regular split-FP system). But perhaps I miss something here. So it would help to better describe the specific advantage of having a split FRCaMP protein-protein interaction sensing system over a regular split FP fluorescence complementation system. To use the split system as calcium sensor I think is not advised, but in the text this is not clear.
- 3) At the end of the introduction (line 51-61) many specific results are summed up, and I think that piece is redundant with the abstract and with the conclusions. I would recommend to shorten this part significantly and to only describe the main features of the new sensor in general terms (not to give numbers).
- 4) In line 212 aggregates are mentioned. As correctly described above that line, this corresponds to lysosomal accumulation and degradation of this RFP-probe. I think in this respect the term aggregates is misleading because it can be confused with non-specific and biologically problematic protein aggregation in the cytosol. The lysosomal-autophagy related accumulation on the other hand is reflecting a normal physiological process of degrading excess of ectopically expressed cytoplasmic proteins and in my opinion should be regarded as a healthy sign, especially for neurons that are more vulnerable cells in terms of protein aggregation (for instance in Alzheimers, Parkinsons, Huntington and BSE disease), see for instance Annu Rev Cell Dev Biol. 2018 Oct 6; 34: 545–568.
Minor issues:
Line 24: purified -> the purified
Line 71: circularity -> the circularity
Line 30: impact -> the impact
Figure 1C please change color of cyan curve to darker color (e.g. dark blue or green) so that it is more clear against a white background
Line 280: It could be better explained why this result indicates no interaction with the cellular environment.
Line 468 suggestion -> The suggestion.
Author Response
Response to Reviewer 1 Comments
We thank reviewer 1 for his/her review, valuable comments and useful suggestions, which we have addressed entirely in the revised manuscript.
Reviewer #1:
The paper by Subach et al describes the evolution and characterization of a new red fluorescent calcium sensor FRCaMP that incorporates a calcium sensing domain originating from fungus.
Overall, the paper is clear and convincing in describing the evolution and characterization of the new red calcium sensor. Because many aspects are quantified also a good comparison can be made to existing state of the art.
There are four points of attention:
Point 1: ) In the evolution, not only new calcium sensing domains (such as CaM and M13) and linkers were included, but also the circularly permutated mApple RFP was mutagenized. The rationale for the latter is not explained. In the evolution many amino acids of the cpApple were altered. Although the main spectral properties of the cpRFP in the sensor are not changed, the photostability seems to be much decreased. Which seems to be the only drawback of the FRCaMP sensor. The question is whether this can be pinpointed to certain amino acids in the cpApple part, by looking at the diverse mutants that they isolated during evolution? The other question is whether this decreased photostability is also partially explained by photochromism. For instance, it is known that mApple is strongly photochromic (quenched by yellow light and restored with blue light). Hence the question here is to what extent the decreased photostability of FRCaMP vs RCaMP is explained by reversible photochromism and/or irreversible photobleaching ? And to what extent can it be attributed to the mutations found in the cpApple RFP?
Response 1: The mutation found in cpmApple part were not inserted by rational mutagenesis but appeared during random mutagenesis. Since we did not characterize the diverse mutants appeared during evolution for photostability and amino acids sequences, so we can not convincingly pinpoint the certain amino acids in the cpmApple part, which affected photostability of the FRCaMP indicator. However, since the side chains of the amino acids for two mutations (K262E and X269S) were directed inside the β-barrel of the FRCaMP indicator, so we added in the Main text, p. 3, that ”We speculate that K262E and X269S mutations can somehow affect the photostability of FRCaMP (as demonstrated below) and these mutations can be considered as the main targets for directed mutagenesis of the FRCaMP indicator in order to improve its photostability; indeed, position 269 was responsible for reversibly photoswitchable-like phenotype in such red fluorescent proteins as rsTagRFP, rsCherry, and KFP1 [9]. “
In the revised manuscript, page 5 we added: “mApple0.5 protein revealed notable photochromism [10]. To address the question of whether photochromism affects the photostability of FRCaMP compared to R-GECO1, we photobleached FRCaM and R-GECO1 GECIs in the sat-state using the mercury lamp described above but with 30-second periods of darkness between the photobleaching cycles (Figure S3). In contrast to R-GECO1, under these conditions FRCaMP demonstrated notable photochromism (Figure S3). Thus, photochromism decreases the one-photon photostability of the FRCaMP indicator.”
Point 2: The rationale for developing the split sensor (fig 7) is unclear. First it seems that the negative control designed to not show an interaction does show fluorescence (albeit 3.5 times lower, see fig 7c) and hence the question is: how specific is this? To screen for an unknown interaction this split system would yield many false positives. Furthermore, it seems not to be advantageous to need to use Ionomycin to read out the protein-protein interaction (as compared to a regular split-FP system). But perhaps I miss something here. So it would help to better describe the specific advantage of having a split FRCaMP protein-protein interaction sensing system over a regular split FP fluorescence complementation system. To use the split system as calcium sensor I think is not advised, but in the text this is not clear.
Response 2: The specificity of BiFC with regular EYFP- and Venus-based fragments was determined as 3-5- and 4-10-fold increase of brightness derived from wild-type bFos as compared to those derived from bFOSDZip within the 125-500 ng range of amount of plasmids in 12-well format, respectively (Y. John Shyu, et al. 2006, BioTechniques, v. 40: 61-66); showing larger specificity of BiFC signal at lower plasmids concentration used for transfection. Hence, even optimized Venus-derived split BiFC system demonstrates non-specific signal in case of bFos/bJun pair meaning that there is fluorescence in the negative control bFosDZip/bJun designed to not show an interaction. In the case of FRCaMP we used 1600 ng concentration of plasmids in 24-well format which is approximately 6-fold higher as compared with concentration used in the paper of Y. John Shyu, et al. 2006, BioTechniques, v. 40: 61-66. Even under these unfavorable conditions split-version of FRCaMP demonstrated specificity of 2.3-3.5-fold.
In this respect, in the revised manuscript, Legend to the Figure 7, we added information about plasmids concentration: “We used 1600 ng concentration of the plasmids for the transfection in 24-well format.”
In the revised manuscript, Main text, page 12, we added “At low physiological calcium ions concentrations, bJun-FRCaMPC/bFos-FRCaMPN pair demonstrated 2.3-fold (p < 0.0001) larger averaged brightness compared to the control bJun-FRCaMPC/bFOSDZip pair (Figure S6).” In the revised manuscript, Supplementary Information, we also added Figure S6 which demonstrates statistically significant difference between interacting and non-interacting proteins at low physiological calcium concentration in the nuclei of the HeLa cells for split-version of FRCaMP. Hence, Ionomycin is not necessary to read out protein-protein interactions.
In the revised manuscript, Main text, page 12, we added that “Overall, in the specific case described above, the spit FRCaMP-based protein-protein interactions sensing system additionally allows the detecting calcium ions elevation in the same fluorescence channel, suggesting that for this reason it has a particular advantage over the regular split FP-based fluorescence complementation system; however, in general case, we don’t recommend using FRCaMP-based split system for detecting protein-protein interactions due to heterodimerization between M13-like peptide and CaM at the elevated calcium ions concentrations, which can lead to incorrect conclusions about protein-protein interactions.”
Finally, we speculated in Conclusions section that the results obtained in this paper open possibility to use split-FRCaMP per se (using M13-like peptide and calmodulin pair interacting or heterodimerizing in the presence of elevated calcium concentrations) for acquiring of integrated information about calcium elevations in neurons similar to the CAMPARI calcium integrator but without the need for illumination of the neurons with 405 nm light. Further experiments are needed to confirm or disprove this suggestion.
Point 3: At the end of the introduction (line 51-61) many specific results are summed up, and I think that piece is redundant with the abstract and with the conclusions. I would recommend to shorten this part significantly and to only describe the main features of the new sensor in general terms (not to give numbers).
Response 3: In the revised manuscript, Main text, at the end of introduction, we shortened this part and described the main features of the FRCaMP in general terms and deleted the numbers.
Point 4: In line 212 aggregates are mentioned. As correctly described above that line, this corresponds to lysosomal accumulation and degradation of this RFP-probe. I think in this respect the term aggregates is misleading because it can be confused with non-specific and biologically problematic protein aggregation in the cytosol. The lysosomal-autophagy related accumulation on the other hand is reflecting a normal physiological process of degrading excess of ectopically expressed cytoplasmic proteins and in my opinion should be regarded as a healthy sign, especially for neurons that are more vulnerable cells in terms of protein aggregation (for instance in Alzheimers, Parkinsons, Huntington and BSE disease), see for instance Annu Rev Cell Dev Biol. 2018 Oct 6; 34: 545–568.
Response 4: In the revised manuscript, Main text, we replaced term aggregates with “puncta-like structures” or “puncta”.
In the revised manuscript, Main text, page 9, we also added: “The lysosomal-autophagy related accumulation is reflecting a normal physiological process of degrading excess of ectopically expressed cytoplasmic proteins and might be regarded as a healthy sign, especially for neurons that are more vulnerable cells in terms of protein aggregation (for instance in Alzheimers, Parkinsons, Huntington and bovine spongiform encephalopathy (BSE) diseases) [14].”
Minor issues:
Point 5: Line 24: purified -> the purified
Response 5: In the revised manuscript, Main text, Line 25, the English editing service corrected “purified protein” to “the purified protein”.
Point 6: Line 71: circularity -> the circularity
Response 6: In the revised manuscript, Main text, Line 79, the English editing service corrected “Circularly permuted” to “A circularly permuted”.
Point 7: Line 30: impact -> the impact
Response 7: In the revised manuscript, Main text, Lines 528-529, the English editing service corrected “possible impacts” to “the possible impacts”.
Point 8: Figure 1C please change color of cyan curve to darker color (e.g. dark blue or green) so that it is more clear against a white background
Response 8: In the revised manuscript, Figure 2c, we changed color of cyan curve to blue one.
Point 9: Line 280: It could be better explained why this result indicates no interaction with the cellular environment.
Response 9: In the revised manuscript, Main text, page 11, Lines 306-307 and 315-316 to better explain why this result indicates no interaction with cellular environment, we added “This result means that to respond to calcium ions the CaM needs to form complex with the M13-like peptide in both FRCaMP and GCaMP6s.” and “These results suggest that the CaM in the truncated FRCaM protein does not form a complex with an intracellular protein having an amino acids sequence similar to M13-like peptide or other proteins, and therefore does not interact with the intracellular environment unlike the CaM in the truncated GCaM6s protein.”.
Point 10: Line 468 suggestion -> The suggestion.
Response 10: In the revised manuscript, Main text, Line 539, the English editing service changed this sentence.
To additionally address points from 5 to 7 and 10, we sent manuscript to the English editing service.

Reviewer 2 Report
The authors presented development of a new red GECI with similar or improved properties over known GECI's evaluated. The construct, combining M13 peptide, cpmApple red fluorescent protein, and CaM from S. pombe fungus, was modified for enhanced GECI capabilities and reduced interaction with environmental species, such as calcium ions. This was a well considered design approach, with a thorough analysis, and conclusions consistent with the results.
I had only a few minor comments.
First, the manuscripts would be improved with minor editing for language.
Second, the authors note on p. 2 that optimal activity requires Kd in the range 100-200nM for neuronal activity. However, FRCaMP described on p. 5, in the presence of 1 mM Mg2+ ions, had a Kd around 441 nM, or 214 nM in the absence of Mg2+ ions. Since the presence of Mg2+ would be observed in the cellular environment, was the Kd listed on p. 2 for the presence or absence of Mg2+?
Author Response
Response to Reviewer 2 Comments
We thank reviewer 2 for his/her review, valuable comments and useful suggestions, which we have addressed entirely in the revised manuscript.
Reviewer #2:
The authors presented development of a new red GECI with similar or improved properties over known GECI's evaluated. The construct, combining M13 peptide, cpmApple red fluorescent protein, and CaM from S. pombe fungus, was modified for enhanced GECI capabilities and reduced interaction with environmental species, such as calcium ions. This was a well considered design approach, with a thorough analysis, and conclusions consistent with the results.
I had only a few minor comments.
Point 1: First, the manuscripts would be improved with minor editing for language.
Response 1: To improve English language we submitted manuscript to the English Editing service.
Point 2: Second, the authors note on p. 2 that optimal activity requires Kd in the range 100-200nM for neuronal activity. However, FRCaMP described on p. 5, in the presence of 1 mM Mg2+ ions, had a Kd around 441 nM, or 214 nM in the absence of Mg2+ ions. Since the presence of Mg2+ would be observed in the cellular environment, was the Kd listed on p. 2 for the presence or absence of Mg2+?
Response 2: In the revised manuscript, Main text, page 2, we emphasized that optimal affinity to calcium ions is considered in the absence of magnesium ions, i.e. we replaced “To adjust its calcium affinity to ~100-200 nM value which is optimal for visualization of neuronal activity, …” with “To adjust its calcium affinity (in the absence of Mg2+ ions) to ~100-200 nM value which is optimal for visualization of neuronal activity, …”

Reviewer 3 Report
In this manuscript, the authors developed and characterized a red fluorescent genetically encoded calcium indicator. The GECI showed faster decay rate compared to NCaMP7 and did not interact with cytosolic environment. Lastly, the authors demonstrated a potential split-version that allowed simultaneous detection of calcium transient and protein interaction. Overall, I find the topic particularly interesting. Yet, the manuscript can benefit of improvement given the aurhos address the following:
Major
- The authors have characterized the impact of pH on the dynamic range of FRCaMP and have used buffer solutions for both in vitro experiments and cell studies. However, the temperature could also affect the Kd of calcium indicator. The authors need to clarify if temperature is controlled in all the experiments.
- In table 1, the authors compared the in vitro properties between FRCaMP and R-GECO1. The authors observed that FRCaMP had lower dF/F values compared to R-GECO1. However, based on the information in the table, the FRCaMP has higher sat- brightness (36% compare to EGFP) and lower apo- brightness (2.5% compare to EGFP) than R-GECO1 (30% sat- brightness and 2.7% apo- brightness). Can the authors clarify that the data are presented correctly?
- In figure 3c, the authors compared the dF/F of FRCaMP and R-GECO1 in HeLa cells. The authors have 10 cells in FRCaMP group and 36 cells in R-GECO1 group. Was all the cells from 1 replicate or multiple independent experiments? Why there are less cells in FRCaMP group?
- The authors monitored the spontaneous neuronal activity with cells co-expressing FRCaMP and green NCaMP7. The authors observed similar rise time and faster decay half-times for FRCaMP. The authors should include a figure and define how the rise and decay time were quantified. Also, the authors claimed that the average dF/F response between FRCaMP and NCaMP7 are similar. However, figure 4a showed that the FRCaMP had higher peak intensity and more details than NCaMP7 likely due to the faster decay rate of FRCaMP. If this is the case, the average dF/F might not be an accurate parameter to compare.
- The authors compared the response of FRCaMP, R-GFCO1, NCaMP7 and GCaMP6s in induced neuronal activity. However, the cells only co-expressed FRCaMP and NCaMP7. How were the data of R-GFCO1and GCaMP6s acquired? Was it all from previous publications? If so, how were the statistical analysis done?
- The authors claim that that “both calcium transients and protein-protein interactions in the cytosol” was visualized. However, the increase of fluorescent intensity requires both calcium influx and bJun-FRCaMPC/bFos-FRCaMPN protein binding. The authors did not show enough data that the “calcium transient” and “protein-protein interactions” can be visualized independently. The authors need to provide more evidence to support this claim.
- The authors lack details on how the fluorescent intensity was quantified. In figure 4a and 5a, the acquired images showed noticeable background. Did the authors subtract the background? How the background noise was determined? And for the evaluation of intensity values, were the intensity calculated from the entire cell or a sub-region of the cell? These are critical information.
Minor:
- For figure 5c. please present the raw data for as R-GFCO1, NCaMP7 and GCaMP6s in a similar format of FRCaMP. Also, please provide R2 values for each fitting so that the goodness of the fitting can be evaluated.
- Please show zoom in images of figure 7a to show the details of the uneven distribution in the nuclei of Hela cells. Also, please specify if the analysis in section 2.7 was done on cell nuclei, a sub-region of cell or the entire cell.
- The English need some polishing to better convey the information.
Author Response
Response to Reviewer 3 Comments
We thank reviewer 3 for his/her review, valuable comments and useful suggestions, which we have addressed entirely in the revised manuscript.
Reviewer #3:
In this manuscript, the authors developed and characterized a red fluorescent genetically encoded calcium indicator. The GECI showed faster decay rate compared to NCaMP7 and did not interact with cytosolic environment. Lastly, the authors demonstrated a potential split-version that allowed simultaneous detection of calcium transient and protein interaction. Overall, I find the topic particularly interesting. Yet, the manuscript can benefit of improvement given the aurhos address the following:
Major
Point 1: The authors have characterized the impact of pH on the dynamic range of FRCaMP and have used buffer solutions for both in vitro experiments and cell studies. However, the temperature could also affect the Kd of calcium indicator. The authors need to clarify if temperature is controlled in all the experiments.
Response 1: In the revised manuscript, 3.2. Proteins purification and characterization section, we added that “All experiments for protein characterization were performed at room temperature (r.t.).”
In the revised manuscript, 3.4. Mammalian live-cell imaging section we added that “Then, 20 mM HEPES, pH 7.40, was added before imaging and cells were kept at r.t during experiment.”
In the revised manuscript, 3.5. Imaging in primary mouse neuronal cultures section we added that "Cells were imaged using an Andor XDi Technology Revolution multi-point confocal system on DIV 15 (spontaneous activity at 37 °C, 5% carbon dioxide) and 21-22 (electrical field stimulation at r.t.)."
Point 2: In table 1, the authors compared the in vitro properties between FRCaMP and R-GECO1. The authors observed that FRCaMP had lower dF/F values compared to R-GECO1. However, based on the information in the table, the FRCaMP has higher sat- brightness (36% compare to EGFP) and lower apo- brightness (2.5% compare to EGFP) than R-GECO1 (30% sat- brightness and 2.7% apo- brightness). Can the authors clarify that the data are presented correctly?
Response 2: In the original manuscript, Table 1, we wrote that used data for R-GECO1 from original publication (Zhao, Y. et al. Science 2011, 333, (6051), 1888-91), except those data that were marked with an asterisk in the Table 1. The noted difference between DF/F values calculated according to the molecular brightness and protein dilutions on Plate Reader can be attributed to the different buffers, protein concentrations and excitation/emission wavelengths used; as described in “3.2. Proteins purification and characterization” section, for molecular brightness measurements we used concentrated protein in 30 mM HEPES, pH 7.2, 100 mM KCl (buffer A) supplemented with 5 mM CaCl2, or 10 mM EDTA and estimated brightness at the absorption maxima and integrated emission; for DF/F estimation according to the protein dilution to 50 nM concentration on Plate Reader we used buffer 30 mM 3-(N-morpholino)propanesulfonic acid (MOPS), pH 7.2, 100 mM KCl (buffer B) containing 10 mM EGTA or 10 mM Ca-EGTA and Ex525nm/Em580-640nm. So, we clarify that the data are presented correctly.
Point 3: In figure 3c, the authors compared the dF/F of FRCaMP and R-GECO1 in HeLa cells. The authors have 10 cells in FRCaMP group and 36 cells in R-GECO1 group. Was all the cells from 1 replicate or multiple independent experiments? Why there are less cells in FRCaMP group?
Response 3: In the revised manuscript, Figure to the legend 3, we added information that the cells for FRCaMP and R-GECO1 were from two and seven cultures, respectively.
Point 4: The authors monitored the spontaneous neuronal activity with cells co-expressing FRCaMP and green NCaMP7. The authors observed similar rise time and faster decay half-times for FRCaMP. The authors should include a figure and define how the rise and decay time were quantified. Also, the authors claimed that the average dF/F response between FRCaMP and NCaMP7 are similar. However, figure 4a showed that the FRCaMP had higher peak intensity and more details than NCaMP7 likely due to the faster decay rate of FRCaMP. If this is the case, the average dF/F might not be an accurate parameter to compare.
Response 4: In the revised manuscript, Main text, 3.5. Imaging in primary mouse neuronal cultures section, we added: “The rise and decay half-times were calculated as time difference between time point corresponding to the calcium spike maximum and time points at half-maximum at the left and right edges of the spike, respectively.”
Figure 4a shows an example of a calcium trace for only one cell; in this particular case the FRCaMP has higher DF/F values. However, to calculate the averaged DF/F values, we used 16 cells for analysis, this explains the noted discrepancy. We see 1.7-fold difference between FRCaMP and NCaMP7 indicators in averaged DF/F values but this difference is not statistically significant because of large variation in DF/F amplitude for calcium transients during the spontaneous neuronal activity. During electrical stimulation of the neuronal cultures, we see the same 1.7-fold difference in DF/F values between FRCaMP and NCaMP7 indicators but it is statistically significant because of the more reproducible conditions.
Point 5: The authors compared the response of FRCaMP, R-GFCO1, NCaMP7 and GCaMP6s in induced neuronal activity. However, the cells only co-expressed FRCaMP and NCaMP7. How were the data of R-GFCO1and GCaMP6s acquired? Was it all from previous publications? If so, how were the statistical analysis done?
Response 5: In the original manuscript, Legend to the Figure 5, we mentioned that “The dependences of DF/F responses on APs for R-GECO1 and G-CaMP6s were added to compare the results with previous work Barykina, N. V. et. al. Int J Mol Sci 2020, 21, (8).” So, yes, the data for R-GECO1 and GCaMP6s were used from paper published earlier by our group. Statistical analysis was done using the Mann-Whitney Rank Sum Test as described in Statistics section.
Point 6: The authors claim that that “both calcium transients and protein-protein interactions in the cytosol” was visualized. However, the increase of fluorescent intensity requires both calcium influx and bJun-FRCaMPC/bFos-FRCaMPN protein binding. The authors did not show enough data that the “calcium transient” and “protein-protein interactions” can be visualized independently. The authors need to provide more evidence to support this claim.
Response 6: The specificity of BiFC with EYFP- and Venus-based fragments was determined as 3-5-fold and 4-10-fold increase of brightness derived from wild-type bFos as compared to those derived from bFOSDZip within the 125-500 ng range of amount of plasmids in 12-well format (Y. John Shyu, et al. 2006, BioTechniques, v. 40: 61-66); showing larger specificity of BiFC signal at lower plasmids concentration used for transfection. In the case of FRCaMP we used 1600 ng concentration of plasmids in 24-well format which is higher as compared with concentration used in the paper of Y. John Shyu, et al. 2006, BioTechniques, v. 40: 61-66.
In the revised manuscript, Main text, page 12, we added “At low physiological calcium ions concentrations, bJun-FRCaMPC/bFos-FRCaMPN pair demonstrated 2.3-fold (p < 0.0001) larger averaged brightness compared to the control bJun-FRCaMPC/bFOSDZip pair (Figure S6).”
In the revised manuscript, Supplementary Information, we also added Figure S6 which demonstrates statistically significant difference between interacting and non-interacting proteins at low physiological calcium concentration in the nuclei of the HeLa cells for split-version of FRCaMP. Hence, calcium transients and protein-protein interactions can be visualized independently.
Point 7: The authors lack details on how the fluorescent intensity was quantified. In figure 4a and 5a, the acquired images showed noticeable background. Did the authors subtract the background? How the background noise was determined? And for the evaluation of intensity values, were the intensity calculated from the entire cell or a sub-region of the cell? These are critical information.
Response 7: In the revised manuscript, 3.4. Mammalian live-cell imaging section, we added: “To quantify the fluorescence intensity, the background noise determined from the adjacent cell-free area was subtracted from mean fluorescence intensity value for cytosolic (Figure 3, 6) or nuclei (Figure 7) sub-region of the cell of the similar area.”
In the revised manuscript, 3.5. Imaging in primary mouse neuronal cultures section, we added: “To quantify the fluorescence intensity, the background noise determined from the adjacent cell-free area was subtracted from mean fluorescence intensity value for the cytosolic (Figure 4 and 5) sub-region of the cell of the similar area.”
Minor
Point 8: For figure 5c. please present the raw data for as R-GFCO1, NCaMP7 and GCaMP6s in a similar format of FRCaMP. Also, please provide R2 values for each fitting so that the goodness of the fitting can be evaluated.
Response 8: In the revised manuscript, Legend to the Figure 5c, we added: “…(linear fitting for FRCaMP had R² = 0.7006)…”. Since NCaMP7 was used as a reference indicator for calculation of APs numbers (assuming linearity of its response in the examined APs range and 0.4 per 1 AP (Subach, O. M. et al. Int J Mol Sci 2020, 21, (5).), we could not estimate R2 values for NCaMP7.
Since the dependences of DF/F responses vs APs for R-GECO1 and G-CaMP6s were published before and were added in this paper for the comparison from previous work (Barykina, N. V. et. al. Int J Mol Sci 2020, 21, (8)), we did not provide R2 values in this paper. All data were obtained from this publication as stated in the Legend to the Figure 5c.
Point 9: Please show zoom in images of figure 7a to show the details of the uneven distribution in the nuclei of Hela cells. Also, please specify if the analysis in section 2.7 was done on cell nuclei, a sub-region of cell or the entire cell.
Response 9: In the revised manuscript, section 2.7, we added that the analysis was done in the cell nuclei, i.e. we added text highlighted with blue color: “We next co-expressed a bJun-FRCaMPC/bFos-FRCaMPN hetero-dimerizing pair and control bJun-FRCaMPC/bFOSDZip non-dimerizing pair in HeLa cells and compared their DF/F responses to ionomycin-induced calcium transients and brightness at elevated calcium concentrations in the nuclei of the cells.”
In the revised manuscript, Supplementary information, Figure S5, we added zoomed images of the cells selected on Figure 7a.
In the revised manuscript, Legend to the Figure 7, we added text highlighted with blue color: “For the selected cells (marked with label 1), zoomed images and time-lapses are shown in Figure S5.”
Point 10: The English need some polishing to better convey the information.
Response 10: To improve the English we send the revised manuscript to the English Editing service.
